# Attitude, self-efficacy, and perceived risk toward seasonal influenza vaccination among primary care physicians in Qatar: A cross-sectional study

Kamran Aziz[1,2☯], Mansoura Salem Ismail[1,2,3☯], Marwa Bibars[ID][4☯*], Nagah Selim[1,5☯], Ayah Mohamed[6☯], Ahmed Sameer Alnuaimi[1☯], Muna Mehdar AlSaadi[1☯]

1 Primary Health Care Corporation, Doha, Qatar, 2 Qatar University, College of Medicine, Doha, Qatar, 3 Family Medicine, Suez Canal University Faculty of Medicine, Ismailia, Egypt, 4 Ministry of Health and Population, Giza, Egypt, 5 Public Health and Preventive Medicine, Cairo University Faculty of Medicine, Cairo, Egypt, 6 University of the Pacific, Stockton, California, United States of America

☯ These authors contributed equally to this work.
* bibars.marwa@gmail.com

## Abstract

Primary care physicians (PCPs) play a critical role in influencing vaccination decisions, both for their patients and for themselves. However, the success of vaccination programs often depends on the attitudes, perceived risks, and self-efficacy Of PCPs. A cross-sectional study examined influenza-related attitude, self-efficacy, and perceived risk among 190 primary health care physicians using an online survey, 46% of participants believed healthcare professionals do not pose influenza transmission risks to patients. Self-efficacy for vaccination was strongly tied to time availability (73.7% agreement) and institutional vaccine provision (78.9%), with social support from colleagues (79.5%) and relatives (68.9%) further influencing adherence. Male physicians (87.5%) and those with ≥5 prior vaccine doses (88.6%) or recent vaccination (87.3%) reported higher self-efficacy, though chronic disease history showed no significant association. Risk perception disparities emerged: 94.2% acknowledged elevated occupational risk during epidemics, yet only 62.1% perceived personal risk. Similarly, 86.3% viewed influenza as dangerous for patients versus 64.2% for themselves. Higher perceived risk scores correlated with chronic disease history (84.5%), prior vaccination (81.1%), recent vaccination (82.8%), and ≥5 vaccine doses (85.0%). Information sources prioritized official health agencies (Ministry of Public Health: 59.5%; WHO/CDC: 56.3%), while traditional media were least utilized (7.9–21.1%). These findings highlight gaps between professional risk acknowledgment and personal risk mitigation, underscoring the need for targeted strategies to address vaccine hesitancy, improve access, and align perceptions with evidence-based practices in healthcare settings.

**Data availability statement:** The primary health care corporation (PHCC) in Qatar has a data acquisition and sharing policy that prohibits data release without an agreement between the relevant parties . a request to share data can be processed by submitting a legitimate request to PHCC, Qatar Contact of institutional Review Borad ( IRB): researchsection@phcc.gov.qa.

**Funding:** The author(s) received no specific funding for this work.

**Competing interests:** The authors have declared that no competing interests exist.

## Introduction

Seasonal influenza is considered a significant public health concern that leads to sizable morbidity, mortality, and healthcare costs worldwide. It is estimated that there are approximately one billion cases of influenza worldwide each year, with 3–5 million resulting in severe cases and 290,000–650,000 respiratory deaths annually [1,2]. Health workers (HWs) are at increased risk of influenza virus infection, which may vary depending on occupation or setting [3]. Infected HWs could increase the risk of nosocomial infection and community spread [4,5]. The seasonal influenza vaccine is one of the most effective preventive measures to reduce the impact of influenza infections. The World Health Organization (WHO) considers HWs to be a priority group for influenza vaccination and has reinforced this position by supporting countries to develop and implement national, seasonal immunization policies for HWs, but the global vaccination rates among healthcare workers remain suboptimal [6]. Healthcare workers need influenza vaccination every year because the dominant strains shift from season to season and vaccineeffectiveness can vary. This repeated need for vaccination, especially in years with modest effectiveness, can feed hesitancy and misconceptions even among staff caring for high-risk patients. Well-run national seasonal programmes—supported by strong hospital systems—can improve coverage by addressing access, supply, confidence, and regulatory hurdles, which also strengthens preparedness for future pandemics [7,8]. Primary care physicians (PCPs) play a critical role in influencing vaccination decisions, both for their patients and for themselves. However, the success of vaccination programs often depends on the attitudes, perceived risks, and self-efficacy of healthcare providers, including PCPs. [9]. PCPs' attitudes toward the seasonal influenza vaccine significantly influence their recommendation and administration practices, with studies showing that healthcare professionals' personal beliefs about vaccines are strong predictors of their vaccination behaviors, and positive attitudes can enhance vaccination rates, while their perception of the risks associated with influenza also plays a crucial role in vaccination uptake [10–12]. Research has shown that healthcare workers who perceive themselves at a higher risk for contracting or transmitting influenza are more likely to get vaccinated. Self-efficacy, or the belief in one's ability to execute a behavior successfully, is another key factor influencing vaccination practices. PCPs with high self-efficacy regarding the influenza vaccine are more likely to recommend it to patients and receive it themselves. Self-efficacy can be shaped by various factors, including training, knowledge of the vaccine's benefits, and previous positive experiences with vaccination. Enhancing PCPs' self-efficacy through education and training can lead to higher vaccine uptake rate [13–15]. This study aims to examine primary care providers' (PCPs) attitudes, perceived risks, and self-efficacy regarding the seasonal influenza vaccine, which are central to improving influenza vaccine uptake and closing the gap between favorable attitudes and actual behavior, and by empowering individuals with the confidence to ultimately reduce the burden of influenza-related disease and death.

## Method of study

An online survey of 450 primary healthcare physicians working in a primary healthcare corporation in Qatar was conducted from October 11, 2021, to October 31, 2021. A list of physicians' emails was received from the operational department of Primary Health Care Corporation (PHCC). An invitation email with an introduction to the study and a questionnaire was sent. Reminders were sent via email and to the physicians' WhatsApp groups.

PHCC serves around three million people from diverse ethnic, cultural, social, economic, and educational backgrounds, which is representative of the whole community in Qatar. Each health care center (HCC) covers a wide geographical region and represents the first level of contact between individuals from the community and the national healthcare system. PHCC provides services including health education, promotion, vaccinations, screening, and management of acute & chronic diseases. There are currently 31 PHCCs divided into three districts and covering the entire country.

A validated questionnaire from Asma et al. [16] was adapted with minor demographic changes, including PHCC work experience. The second section assessed attitude using 13 items, of which 10 measured self-efficacy, five on social influence, and five on personal competence. Perceived risk and perceived severity of risk were measured using five-point Likert questions. Responses were expressed as follows: 1 means strongly agree, 2 means agree, 3 means neutral, 4 means disagree, and 5 means strongly disagree. The percentage reported was for those who answered as agree or strongly agree, which were considered positive items and were awarded a score of one point. The only exception to this was the first item, "I feel that health professionals are not spreading the disease to their patients," in the attitude score, in which disagree and strongly disagree were given positive scores. Whereas neutral, disagree, and strongly disagree were given a negative score and weren't analysed. The score was calculated by summing the scores of only the positive items and multiplying it by 100/maximum count of items, and then the mean score was calculated for all responses. This contributed towards standardizing the interpretation and comparison of different scores.

### Data analysis

The collected data was analyzed using SPSS version 25, and the quality of the data was checked before proceeding to the analysis. This was accomplished by reviewing the completed surveys for accuracy and consistency, ensuring they were submitted within the designated data collection period and that no outliers were present. Quantitative data were summarized in the form of mean and SD while qualitative data was summarized in the form of frequency tables and charts. -Bivariate analysis was then carried out using the independent t-test for two groups and ANOVA for more than two groups to investigate the association between the dependent variables and the background characteristics of physicians. Effect size is assessed by Cohen's d for differences in the mean between two groups, while the Eta squared test was used for the effect of a factor with more than two groups. The level of statistical significance was set at a $p$-value of less than or equal to 0.05.

### Ethics statement

This was an anonymous, online-based survey utilizing implied informed consent. Participants were informed that by submitting the completed questionnaire, a survey link was distributed to all physicians, accompanied by a statement indicating that clicking the link signified voluntary consent to take part in the study. The questionnaire included a cover page outlining the study's aims and objectives, emphasizing that participation was entirely voluntary and that no identifiable information would be collected. All anonymized responses were received directly by the researcher.

### Ethics approval

The Institutional Review Board at Primary Health Care Corporation in Qatar approved this study (Reference number PHCC/DCR/ 2020/06/047)

## Results

The overall response rate in this study was 42.2%; 106 (55.7%) were 45 years old or older, and more than half of the participants were male. About one-third reported a history of Chronic diseases Table 1.

Most participants (96.3%) had been vaccinated against influenza in the past, while 73.3% of the physicians had received an influenza vaccine in the last season Table 2.

Of 190 doctors, the majority (87.9%) believe that health professionals should be vaccinated for the continuity of health services. Despite this, only 37.4% believe that the influenza vaccine should be mandatory for healthcare workers. Notably, nearly 46% of the respondents feel confident that health professionals do not pose a risk of spreading the influenza virus to their patients Table 3.

When asked about their self-efficacy in successfully getting vaccinated, time and availability were the two most important factors. 73.7% agreed that they would be vaccinated if they had enough time, while 78.9% expressed willingness to be vaccinated if the institute provided the vaccine. Perceived social support for influenza vaccination from relatives and colleagues was frequently reported with individual self-efficacy (68.9% & 79.5% respectively), Table 4.

**Table 1. Background characteristics of primary care physicians working in Primary Healthcare Corporation in Qatar (N ==190).**

| Characteristic | N | % |
|---|---|---|
| Age | | |
| ≤44 | 84 | 44.2 |
| ≥ 45 | 106 | 55.7 |
| Female | 78 | 41.1 |
| Male | 112 | 58.9 |
| Duration of PHCC work experience in years | | |
| <8 | 134 | 70.5 |
| ≥8 | 56 | 29.5 |
| History of chronic diseases | | |
| No | 129 | 67.9 |
| Yes | 61 | 32.1 |

**Table 2. History of Influenza vaccine uptake among primary care physicians working in the primary health care corporation in Qatar (*N*=190).**

| | N | % |
|---|---|---|
| Ever taken influenza vaccine | | |
| No | 7 | 3.7 |
| Yes | 183 | 96.3 |
| influenza vaccination last season | | |
| No | 50 | 26.3 |
| Yes | 140 | 73.7 |
| Doses of seasonal influenza vaccine ever received | | |
| Never | 7 | 3.7 |
| (1-2) | 25 | 13.2 |
| (3-4) | 49 | 25.8 |
| ≥5 | 87 | 45.8 |
| Don't remember | 22 | 11.6 |

**Table 3. Attitude of primary care physicians in Qatar towards Influenza vaccination (*N*=190).**

| Those who answered agree and strongly agree | N | % | 95%confidence interval |
|---|---|---|---|
| I feel that health professionals are not spreading the disease to their patients | 87 | 45.8 | (38.8-52.9) |
| I believe that health professionals should be vaccinated for the continuity of health services | 167 | 87.9 | (82.7-92) |
| Influenza vaccine should be mandatory for health professionals | 71 | 37.4 | (30.7-44.4) |

**Table 4. Distribution of factors influencing the Self-efficacy of primary care physicians towards Influenza vaccination (N ==190).**

| | Agree and strongly agree | | 95%confidence interval |
|---|---|---|---|
| statements | N | % | |
| My relatives believe that my vaccination is important | 131 | 68.9 | (62.1 - 75.2) |
| My institute recommends being vaccinated | 185 | 97.4 | (94.3 - 99) |
| My colleagues believe that my vaccination is important | 151 | 79.5 | (73.3 - 84.7) |
| The Ministry of Health recommends vaccination of health professionals | 187 | 98.4 | (95.8 - 99.6) |
| The health authorities I know recommend vaccination | 182 | 95.8 | (92.2 - 98) |
| I would be vaccinated every year if I have enough time | 140 | 73.7 | (67.1 - 79.6) |
| I would be vaccinated if someone reminds me | 126 | 66.3 | (59.4 - 72.7) |
| I would be vaccinated every year if the vaccine is provided in my institute | 150 | 78.9 | (72.7 - 84.3) |
| I would be vaccinated every year if I am rewarded | 64 | 33.7 | (27.3 - 40.6) |
| I would be vaccinated for influenza every year if sufficient knowledge was given | 125 | 65.8 | (58.8 - 72.3) |

Significant positive associations with self-efficacy mean scores were observed among primary care physicians who were male (87.5%) and those who had received five or more influenza vaccinations (88.6%), particularly in the most recent season (87.3%). In contrast, there was no statistically significant relationship between a history of chronic diseases and self-efficacy scores Table 5.

In assessing risk perception, a difference was reported between physicians' acknowledgment of risk and their personal risk perception. While 94.2% agreed that healthcare professionals face the greatest influenza risk during epidemics, only 62.1% believed they were at high risk. This was reflected in perceived severity; 86.3% considered influenza dangerous for patients, versus only 64.2% for themselves Table 6.

The analysis revealed a statistically significant association between physicians' perceived risk mean scores and select background characteristics. Notably, higher mean scores were observed among those with a history of chronic diseases (84.5), prior influenza vaccination (81.1), influenza vaccination in the last season (82.8), and having received five or more seasonal influenza vaccine doses (85.0). Table 7.

Physicians primarily rely on official websites from the Ministry of Public Health (59.50%), WHO, and CDC (56.30%) for influenza vaccine updates. Conversely, newspapers (21.10%), TV, and flu-specific platforms (7.90%) are the least consulted sources (Fig 1).

## Discussion

The study focused on primary care physicians' attitudes, self-efficacy, and perceived risk regarding accepting the seasonal influenza vaccine.

The study revealed that most participants had received an influenza vaccination in the past, while two-thirds of physicians had been vaccinated during the last season. This is per a study done in the US, where family physicians' vaccination rate was up to 87%, attributed to workplace policies and free access on-site for all staff [17].

**Table 5. Relationship between sociodemographic characteristics of primary care physicians and perceived self-efficacy score in Qatar 2021 (*N*=190).**

| Characteristic | Perceived Self-efficacy Score | | | | | | |
| | Range | Mean | SD | SE | N | *p*-Value | Effect size |
|---|---|---|---|---|---|---|---|
| Age (years) | | | | | | | |
| 22-34 | (78-96) | 86.0 | 5.4 | 1.64 | 11 | 0.91 | 0.003 |
| 35-44 | (58-110) | 85.0 | 11.9 | 1.39 | 73 | | |
| 45-54 | (68-106) | 86.3 | 10.0 | 1.16 | 74 | | |
| ≥55 | (56-110) | 85.3 | 11.6 | 2.05 | 32 | | |
| Gender | | | | | | | |
| Female | (58-110) | 83.0 | 11.0 | 1.25 | 78 | 0.005 | −0.424 |
| Male | (56-110) | 87.5 | 10.3 | 0.97 | 112 | | |
| Duration of PHCC work experience in years | | | | | | | |
| <4 | (68-110) | 85.8 | 10.0 | 1.31 | 58 | 0.55 | 0.011 |
| 4-7 | (56-108) | 85.4 | 11.6 | 1.33 | 76 | | |
| 8-10 | (68-110) | 88.8 | 10.6 | 2.51 | 18 | | |
| ≥11 | (62-108) | 84.3 | 10.5 | 1.70 | 38 | | |
| History of chronic disease | | | | | | | |
| No | (58-110) | 85.4 | 11.1 | 0.98 | 129 | 0.7 | −0.058 |
| Yes | (56-104) | 86.1 | 10.2 | 1.31 | 61 | | |
| Ever taken the influenza vaccine | | | | | | | |
| yes | (58-84) | 76.3 | 9.7 | 3.66 | 7 | 0.038 | −0.912 |
| no | (56-110) | 86.0 | 10.7 | 0.79 | 183 | | |
| Influenza vaccination last season | | | | | | | |
| no | (58-110) | 81.0 | 11.3 | 1.60 | 50 | <0.001 | −0.598 |
| yes | (56-110) | 87.3 | 10.1 | 0.86 | 140 | | |
| Doses of seasonal influenza vaccine received | | | | | | | |
| Never | (58-84) | 76.3 | 9.7 | 3.66 | 7 | <0.001 | 0.098 |
| (1-2) | (58-102) | 80.4 | 11.0 | 2.19 | 25 | | |
| (3-4) | (68-106) | 85.0 | 9.0 | 1.29 | 49 | | |
| ≥5 | (56-110) | 88.6 | 11.1 | 1.19 | 87 | | |

**Table 6. Seasonal influenza vaccine perceived risk and severity of risk among primary care physicians (*N*=190).**

| | N | % | 95%confidence interval |
|---|---|---|---|
| Perceived risk | | | |
| I'm at high risk for influenza | 118 | 62.1 | (55.1 - 68.8) |
| I can spread infection to my patients even if I am asymptomatic | 153 | 80.5 | (74.5 - 85.7) |
| Health professionals are under the highest risk in case of an epidemic | 179 | 94.2 | (90.2 - 96.9) |
| I can spread infection to my family even if I am asymptomatic | 170 | 89.5 | (84.5 - 93.2) |
| Perceived severity of risk | | | |
| Influenza is dangerous for me | 122 | 64.2 | (57.2 - 70.8) |
| Influenza is dangerous for my patients | 164 | 86.3 | (80.9 - 90.6) |
| Influenza is dangerous for my family | 148 | 77.9 | (71.6 - 83.3) |

**Table 7. Relationship between sociodemographic characteristics of primary care physicians and perceived risk score in Qatar 2021 (*N* = 190).**

| Characteristic | Range | Mean | SD | SE | N | p-Value | Effect size |
|---|---|---|---|---|---|---|---|
| Age (years) | | | | | | 0.18] | 0.026 |
| 22-34 | (40 - 100) | 75.6 | 15.3 | 4.61 | 11 | | |
| 35-44 | (49 - 100) | 12.8 | 12.8 | 1.5 | 73 | | |
| 45-54 | (46 - 100) | 82.5 | 12.4 | 1.44 | 74 | | |
| ≥ 55 | (60 - 100) | 82.1 | 11.5 | 2.03 | 32 | | |
| Gender | | | | | | 0.24 | −0.176 |
| Female | (40 - 100) | 79.4 | 13.2 | 1.49 | 78 | | |
| Male | (46 - 100) | 81.6 | 12.2 | 1.16 | 112 | | |
| Duration of PHCC work experience in years | | | | | | 0.45] | 0.014 |
| <4 | (40 - 100) | 78.9 | 13.1 | 1.73 | 58 | | |
| 4-7 | (46 - 100) | 80.8 | 12.7 | 1.46 | 76 | | |
| 8-10 | (57 - 100) | 84 | 12.5 | 2.94 | 18 | | |
| ≥ 11 | (60 - 100) | 81.8 | 11.7 | 1.9 | 38 | | |
| Having chronic diseases | | | | | | 0.70 | −0.058 |
| no | (40 - 100) | 78.9 | 13.2 | 1.16 | 129 | | |
| yes | (63 - 100) | 84.5 | 10.5 | 1.35 | 61 | | |
| ever taken influenza vaccine | | | | | | <0.001 | −0.912 |
| no | (63 - 77) | 71 | 6.5 | 2.45 | 7 | | |
| yes | (40 - 100) | 81.1 | 12.7 | 0.94 | 183 | | |
| history of influenza vaccination last season | | | | | | <0.001 | −0.640 |
| no | (40 - 100) | 75 | 13.5 | 1.91 | 50 | | |
| yes | (46 - 100) | 82.8 | 11.7 | 0.99 | 140 | | |
| Count of seasonal influenza vaccine doses received | | | | | | <0.001 | 0.134 |
| Never | (63 - 77) | 71 | 6.5 | 2.45 | 7 | | |
| (1-2) | (40 - 94) | 72.5 | 14.4 | 2.88 | 25 | | |
| (3-4) | (51 - 100) | 79.2 | 12.7 | 1.81 | 49 | | |
| ≥ 5 | (51 - 100) | 85 | 10.7 | 1.15 | 87 | | |

Most physicians believe that healthcare professionals should be vaccinated to ensure the continuity of health services. Studies have shown that physicians know their occupational obligations towards vaccination [18,19].

According to Polan et al., when healthcare professionals are vaccinated against seasonal influenza, they contribute to the healthcare system's effective response to influenza epidemics [20].

About one-third of the physicians thought the influenza vaccine should be mandatory for healthcare professionals. According to Winston et al., mandatory vaccination increases vaccination rates, but healthcare professionals negatively perceive it [21].

Approximately half of the respondents felt that health professionals do not contribute to the spread of the influenza virus to their patients. However, some studies suggest that the disease is often asymptomatic, and healthcare professionals can spread the influenza virus to their patients and families [22–24].

The availability of the influenza vaccine and the time needed to get vaccinated were crucial factors influencing individuals' self-efficacy regarding vaccine uptake. Approximately two-thirds of participants indicated that they would be willing to get vaccinated if they had enough time and access to the vaccine at their institution. AlMarzooqi et al. reported that one of the frequently cited reasons for not taking the influenza vaccine was a lack of time to take the vaccine (28.9%).

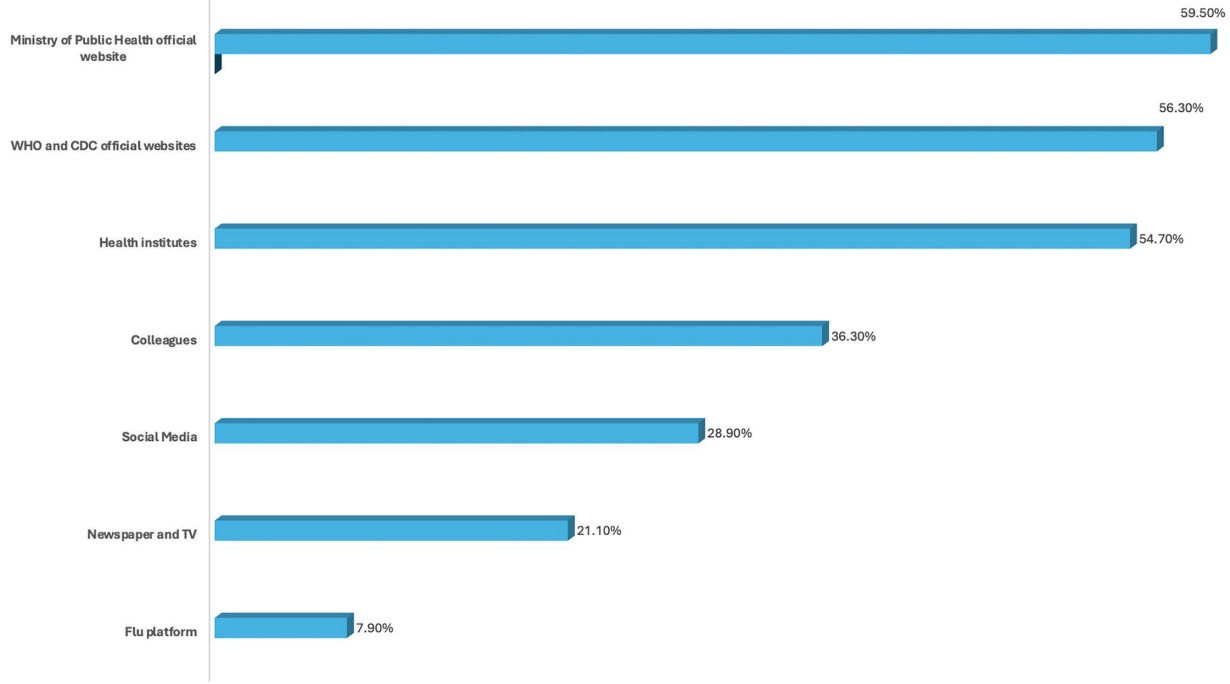

**Fig 1. Bar chart for the source of information.**

The majority (85.1%) of healthcare professionals received the vaccine from the governmental sector, which was also their workplace [25].

According to a study by Cowan et al. in the US, a common reason for not receiving the vaccine was the lack of time, reported by 34% of participants. Similarly, Wallace et al. found that 5–60% of health care workers cited time constraints as a reason for rejecting the influenza vaccine [17,26].

Additionally, social support from family and colleagues plays a significant role in enhancing a person's self-efficacy. Takayanagi et al. showed that "believing that most departmental colleagues had been vaccinated" (P < .0001) was significantly associated with compliance with influenza vaccination [24].

Among male primary care physicians, those who had received five or more influenza vaccinations in the past showed a significant positive association with self-efficacy. This finding aligns with the results of Alenazi et al., who found that increasing age, longer work duration in health services, being male, and being a physician are associated with significantly larger vaccination compliance (p = 0.02, p = 0.07, p = 0.01, p = 0.01), respectively [27].

The survey found no statistically significant relationship between having a history of chronic diseases and self-efficacy. This is contrary to a study by AlMarzooqi et al., where it was noted that participants with diabetes, bronchial asthma, and obesity had increased acceptance of the influenza vaccine (P < .001) [25].

There is a notable difference in how physicians acknowledge risk versus their perception of risk. Most physicians agree that healthcare professionals are at the highest risk for influenza during epidemics; however, fewer believe they are at high risk. Studies have found that one main reason for not receiving the seasonal influenza vaccine among healthcare professionals was that they did not recognize themselves as at risk [28,29].

According to AlMarzooqi et al., the most common reason among health care professionals for not being vaccinated was their belief that they were not at high risk of contracting influenza [25].

The study revealed that physicians primarily rely on official sources for information about the influenza vaccine, such as the Ministry of Public Health (59.5%), WHO, and CDC (56.3%), which is slightly lower than the 70.5% reported by AlMarzooqi et al. in 2018 [25]. In contrast, they consult electronic and print media and flu-specific resources much less frequently for influenza information. Alame et al. reported that the participants relied on various sources for vaccine recommendations, including international organizations (e.g., WHO) and the Ministry of Health [30].

## Limitations of the study

The low response rate and small sample size may restrict generalizability. Vaccination uptake was self-reported rather than verified with official records, introducing potential social desirability bias. Recall bias is also possible; however, the survey was conducted over three weeks during the flu season (October 2021).

## Conclusion

This study highlights a gap between primary care physicians' recognition of professional influenza risk and their vaccination behaviours. While most support vaccination to maintain healthcare services, fewer perceive themselves at high risk or endorse mandatory policies. Self-efficacy driven by institutional support, time availability, and social reinforcement plays a key role in vaccine uptake. These findings highlight gaps between professional risk acknowledgment and personal risk mitigation, underscoring the need for targeted strategies to address vaccine hesitancy, improve access, and align perceptions with evidence-based practices in healthcare settings.

## Acknowledgments

The authors of this manuscript would like to thank primary care physicians in Qatar who participated in this survey.

## Author contributions

**Conceptualization:** Kamran Aziz, Mansoura salem Ismail.

**Data curation:** Kamran Aziz, Mansoura salem Ismail.

**Formal analysis:** Ahmed Sameer Alnuaimi.

**Methodology:** Mansoura salem Ismail, Marwa Bibars, Ahmed Sameer Alnuaimi.

**Writing – original draft:** Kamran Aziz, Mansoura salem Ismail, Marwa Bibars, Nagah Selim, Ayah Mohamed, Ahmed Sameer Alnuaimi, Muna Mehdar AlSaadi.

**Writing – review & editing:** Kamran Aziz, Mansoura salem Ismail, Marwa Bibars, Nagah Selim, Ayah Mohamed, Ahmed Sameer Alnuaimi, Muna Mehdar AlSaadi.

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
