## [Decision Letter · Decision Letter 0]

6 Jun 2025

PONE-D-25-15876Assessing attitude, self-efficacy, and perceived risk toward seasonal influenza vaccination among primary care physicians in Qatar: A cross-sectional studyPLOS ONE

Dear Dr. Bibars,

Thank you for submitting your manuscript to PLOS ONE. After careful consideration, we feel that it has merit but does not fully meet PLOS ONE’s publication criteria as it currently stands. Therefore, we invite you to submit a revised version of the manuscript that addresses the points raised during the review process.

We look forward to receiving your revised manuscript.

Kind regards,

Ameel Al Shawi

Academic Editor

PLOS ONE

Reviewers' comments:

Reviewer's Responses to Questions

**Comments to the Author**

1. Is the manuscript technically sound, and do the data support the conclusions?

Reviewer #1: Partly

Reviewer #2: Partly

2. Has the statistical analysis been performed appropriately and rigorously? 

Reviewer #1: Yes

Reviewer #2: Yes

3. Have the authors made all data underlying the findings in their manuscript fully available?

Reviewer #1: Yes

Reviewer #2: Yes

4. Is the manuscript presented in an intelligible fashion and written in standard English?

Reviewer #1: Yes

Reviewer #2: No

5. Review Comments to the Author

Reviewer #1: General comments:

Thank you for giving me the opportunity to review this manuscript. Previous studies have primarily focused on primary care doctors' attitudes toward influenza vaccine, with limited research on their self-efficacy. Therefore, this study may contribute further to the literature. However, further details are required to add clarity to the manuscript.

Specific comments as below:

Title:

I suggest removing the word ‘Assessing’ in the title to be more concise.

Introduction:

Line 61: ‘...nearby 0.65 million…’ is not clear. I suggest changing the word ‘nearby’ or rephrase.

Line 69: to add reference(s) supporting the statement ‘vaccination rates among health care workers remain suboptimal.’

The aim of the study is not clearly stated.

Method:

A brief background of primary healthcare corporation in Qatar will be useful for the reader as it is not clear whether this constitutes a chain of clinics.

More details about the questionnaire used in this study are required. Was the questionnaire adopted unchanged from Asma et al or were any modifications made? This is in view that Asma et al’s study was regarding factors affecting influenza vaccine uptake and not directly on attitude. If any translation of language was done, this need to be stated too.

If the questionnaire was adapted, whether pilot testing and validation were conducted.

How many items are there for attitude, self-efficacy and perceived risks respectively?

What is meant by ‘positive items?’

The statement: ‘each positive item was awarded a score of one point’ needs more explanation as it is not clear. How was ‘negative item’ analysed?

Line 101: it is stated that total score was calculated. However, Table 5 and 7 shows mean self-efficacy and perceived risk score. This needs to be clarified.

How were the primary healthcare physicians identified, e.g. was it through a database? How were they invited to the study, e.g. was it via emails?

What was the total number of physicians invited to participate in the study?

Were there any strategies done to maximise the response rate e.g. reminders?

Results:

Tables 1 & 2: to state the total number of respondents (N) after the title (as stated in Table 3). The last row in the table ‘Total’ is not relevant and can be removed.

Table 1: it was not stated what PHCC means.

Table 2: columns 2 & 3, there are no title to the columns, whether it describes number and percentages.

Table 3 and 4: does column ‘n’ represents the number of respondents who answered ‘agree’ and strongly agree’ to each statement (referring to the method of study)? If so, it also needs to be stated in the table.

Line 148: ‘Time’ should be written in lower case.

The items in Table 4 appears to be factors influencing influenza vaccination or factors influencing self-efficacy. Self-efficacy is assessed by determining ‘levels of confidence in performing tasks and whether or not someone can do or accomplish a given task’. This need to relate with the objectives of this study which need to be clearly stated.

Table 5 and Table 7:

The term ‘physicians’ frequency’ is confusing.

Stating the responses as ‘yes’ or ‘no’ for ‘history of chronic disease’, ‘ever taken influenza vaccine’ and ‘influenza vaccination last season’ will give more clarity compared to ‘negative’ and ‘positive’.

Written as ‘Ever taken influenza vaccine’ in Table 5, whereas in Table 7, it is written as ‘Positive history of influenza vaccination’. It is best to standardise the statements if it refers to the same questionnaire statement. Same with ‘influenza vaccination last season’.

Table 7: suggest removing ‘[NS]’.

Discussion:

Discussion can be further elaborated:

Is there a policy on influenza vaccination in the healthcare corporation of Qatar which may explain the why two thirds of the respondents had been vaccinated?

What are the possible reasons that may explain the respondents having not enough time and no access to the vaccine?

To include discussion on limitations of this study and future recommendations.

Lines 212 – 215, 220 – 224, 242 – 243: the flow of writing need to be improved as it appears disjointed.

References:

To check reference (15): the page in the link could not be found. It is not necessary to explain on what the reference is about in the reference list.

Reviewer #2: Dear Author,

Thank you for this study on very important topic of global concerns.

I think you still need a lot of work to put into it for it to fulfil the minimum criteria of a good research article. Please find below my comments:

Ln 1: I don't think this word (Assessing) is needed in the title.

You may leave it if you feel so strong about it.

Ln 36 -57: While you have very important points in this abstract, it does not have the minimum requirement of a good research abstract.

Please re-write this section bearing in mind the following:

-Introduction

-Methods

-Results

-Discussions

-straight to point and brief

-Word count

Note that your Abstract should be able to invite the readers to read the whole of the study. Thanks

Ln 66/67: Reference please.

Ln 67: T

Ln 67:is reinforced

Ln 74/75: Reference please

Ln 77-79: Reference please

Ln 86: replace with "to"

Ln 88- 90: This is the statement of the scientific value of your study. You need to summarize the "hows?" in this introduction as well.

I will prefer the paragraph is framed as:

This research looked into improving attitudes and self-efficacy........... by:

1.Boosting the knowledge of the PCPs in training about influenza dx..

2. Ready availability of the influenza vax to pcp at their working units..

3...

4....

etc.

Ln94-95: consider replacement with "demographic characteristics"

Ln 105/106: How? Please state

Table2: What exactly do you mean by this question? Influenza vaccine is routinely 1 dose of either the trivalent or quadrivalent vaccine administered annually. Please clarify or re-frame your question

Ln 185: Good discussion with adequate comparison with available similar studies. Thank you

Ln 244: While this study addressed a common global seasonal health issue, I think the conduct of this research leaves a lot of factors that may affect its validity and transferability.

Please can you highlight the limitations of your study.

Thanks

Ln 245: This conclusion is too scanty and lacks the basic elements of a good conclusion of a research article.

Please rewrite bearing in mind: restatement of the social value of the study, summary of key arguments and brief discussion of the implications of your research.

Thanks.

6. PLOS authors have the option to publish the peer review history of their article (what does this mean? ). If published, this will include your full peer review and any attached files.

**Do you want your identity to be public for this peer review?** For information about this choice, including consent withdrawal, please see our Privacy Policy .

Reviewer #1: No

Reviewer #2: **Yes: ** Adeloye Amoo Adeniji (MBBS; MMed; FCFP; FACRRM)

---

## [Author Response · Author response to Decision Letter 1]

16 Sep 2025

1. I have ensured that my manuscript meets PLOS ONE's style requirements, including those for file naming.

2. Regarding data availability: The primary health care corporation (PHCC) in Qatar has a data acquisition and sharing policy that prohibits data release without an agreement between the relevant parties. A request to share data can be processed by submitting a legitimate request to PHCC, Qatar. Contact info of Institutional Review Borad ( IRB) : researchsection@phcc.gov.qa

Comments to the Author

1. . Is the manuscript technically sound, and do the data support the conclusions? The conclusion was changed to reflect the data presented.

2. Is the manuscript presented in an intelligible fashion and written in standard English? The manuscript was re-edited.

Reviewer #1:

We sincerely appreciate the reviewer’s thoughtful feedback. In response, we have revised the manuscript to include further details and clarifications aimed at improving clarity. These changes have been highlighted in both the revised manuscript and this rebuttal letter. We believe that the revisions enhance the overall quality of the paper and better emphasize its contribution to the existing literature.

Title:

I suggest removing the word ‘Assessing’ in the title to be more concise .

- It was removed.

Introduction:

Line 61: ‘...nearby 0.65 million…’ is not clear. I suggest changing the word ‘nearby’ or rephrasing.

- It is estimated that there are approximately one billion cases of influenza worldwide each year, with 3–5 million resulting in severe cases and 290,000 to 650,000 respiratory deaths annually [1,2].

Line 69: to add reference(s) supporting the statement ‘vaccination rates among health care workers remain suboptimal.’

- but the global vaccination rates among healthcare workers, remain suboptimal. [6] Dini G., Toletone A., Sticchi L., Orsi A., Bragazzi N.L., Durando P. Influenza vaccination in healthcare workers: A comprehensive critical appraisal of the literature. Hum. Vaccin. Immunother. 2018;14:772–789. doi: 10.1080/21645515.2017.1348442.

The aim of the study is not clearly stated.

- This study aims to examine primary care providers’ (PCPs) attitudes, perceived risks, and self-efficacy regarding the seasonal influenza vaccine.

Method:

A brief background of primary healthcare corporation in Qatar will be useful for the reader as it is not clear whether this constitutes a chain of clinics.

- The primary health care centers (PHCC) serve around three million people from diverse ethnic, cultural, social, economic, and educational backgrounds, which is representative of the whole community in Qatar. Each health center (HC) covers a wide geographical region and represents the first level of contact between individuals from the community and the national healthcare system. PHCC provides services including health education, promotion, vaccinations, screening, and management of acute & chronic diseases. There are currently 31 PHCCs divided into 3 districts and covering the entire country.

More details about the questionnaire used in this study are required. Was the questionnaire adopted unchanged from Asma et al or were any modifications made? This is in view that Asma et al’s study was regarding factors affecting influenza vaccine uptake and not directly on attitude. If any translation of language was done, this needs to be stated too.

- The questionnaire was obtained from Asma et al. [16] study, which was validated in the English language with a few changes in the demographic characteristics such as working experience in PHC.

If the questionnaire was adapted, whether pilot testing and validation were conducted.

- There was neither pilot testing nor validation.

How many items are there for attitude, self-efficacy, and perceived risks respectively?

- Three statements covering attitude. Self-efficacy represent the major part of attitude items (10 out of 13) therefore we focused on self-efficacy using 10 items which cover social effects, measured by 5 statements, and personal competence measured by 5 statements. Perceived risk and perceived severity of risk were measured using 7 statements. All of which were assessed using five-point Likert questions

What is meant by ‘positive items?’ The statement: ‘each positive item was awarded a score of one point’ needs more explanation as it is not clear. How was ‘negative item’ analysed?

- Responses were expressed as follows: 1 means strongly agree, 2 means agree, 3 means neutral, 4 means disagree, and 5 means strongly disagree.

- The percentage reported was for those who answered as agree or strongly agree , which were considered positive items and were awarded a score of one point. The only exception to this was the first item “ I feel that health professionals are not spreading the disease to their patients” in the attitude score, in which disagree and strongly disagree were given positive scores.

- Whereas neutral, disagree and strongly disagree were given negative score and weren’t analysed.

- The score was calculated by summing the scores of only the positive items and multiplying it by 100/maximum count of items, and then the mean score was calculated for all responses. This contributed towards standardizing the interpretation and comparison of different scores.

Line 101: it is stated that total score was calculated. However, Table 5 and 7 shows mean self-efficacy and perceived risk score. This needs to be clarified.

- The score was calculated by summing the scores of only the positive items and multiplying it by 100/maximum count of items, and then the mean score was calculated for all responses. This contributed towards standardizing the interpretation and comparison of different scores.

How were the primary healthcare physicians identified, e.g. was it through a database? How were they invited to the study, e.g. was it via emails? Were there any strategies done to maximise the response rate e.g. reminders? What was the total number of physicians invited to participate in the study?

- An online survey of 450 primary healthcare physicians working in primary healthcare corporation in Qatar was conducted from October 11, 2021, to October 31, 2021.

- A list of Physicians’ emails was received from the operational department of PHCC. An invitation email with an introduction to the study and a questionnaire was sent. Reminders were sent via email and to the physicians’ WhatsApp groups.

Results:

Tables 1 & 2: to state the total number of respondents (N) after the title (as stated in Table 3). The last row in the table ‘Total’ is not relevant and can be removed.

- Such modifications were made.

Table 1: It was not stated what PHCC means.

- It is now added to the title.

Table 2: columns 2 & 3, there are no title to the columns, whether it describes number and percentages.

- This has been modified.

Table 3 and 4: Does column ‘n’ represent the number of respondents who answered ‘agree’ and strongly agree’ to each statement (referring to the method of study)? If so, it also needs to be stated in the table.

- Yes, it does. It is now stated in the table.

Line 148: ‘Time’ should be written in lower case.

- It was edited.

The items in Table 4 appears to be factors influencing influenza vaccination or factors influencing self-efficacy. Self-efficacy is assessed by determining ‘levels of confidence in performing tasks and whether or not someone can do or accomplish a given task’. This need to relate with the objectives of this study which need to be clearly stated.

- The table title was changed to Table 4: distribution of factors influencing Self-efficacy of primary care physicians towards Influenza vaccination (N =190).

-

Table 5 and Table 7:

The term ‘physicians’ frequency’ is confusing.

- It was changed to “N” in both tables.

Stating the responses as ‘yes’ or ‘no’ for ‘history of chronic disease’, ‘ever taken influenza vaccine’ and ‘influenza vaccination last season’ will give more clarity compared to ‘negative’ and ‘positive’.

- It has been modified to yes/no format.

Written as ‘Ever taken influenza vaccine’ in Table 5, whereas in Table 7, it is written as ‘Positive history of influenza vaccination’. It is best to standardize the statements if it refers to the same questionnaire statement. Same with ‘influenza vaccination last season’.

The question was about “ever taken influenza vaccine (no/yes)

history of influenza vaccination last season (no/yes )

- It has been standardized.

Table 7: suggest removing ‘[NS]’.

- It has been removed.

Discussion:

Discussion can be further elaborated:

Is there a policy on influenza vaccination in the healthcare corporation of Qatar which may explain the why two thirds of the respondents had been vaccinated?

According to the Ministry of Public Health, flu vaccines are made available free of charge every year starting October 1st. The 2024–25 campaignincludes over 80 health facilities, including 31 PHCC centers, Hamad Medical Corporation and outpatient clinics, and private hospitals.

What are the possible reasons that may explain the respondents having not enough time and no access to the vaccine?

- Internal communications and schedules show that physicians often work from 7 AM to 2 PM in PHCC settings, with tightly packed clinical and administrative responsibilities. This leaves little room for preventive health actions as vaccinations unless explicitly scheduled.

To include discussion on the limitations of this study and future recommendations.

o limitations :

Despite Significant findings, the study has some limitations. Vaccination uptake was self-reported and not verified with official records. This could lead to biases such as social desirability bias. There might be a possibility of information recall bias, however, the survey was conducted for three weeks in October 2021 during flu season. The sample size was small, which may limit the generalizability

o Future recommendations

These findings highlight gaps between professional risk acknowledgment and personal risk mitigation, underscoring the need for targeted strategies to address vaccine hesitancy, improve access, and align perceptions with evidence-based practices in healthcare settings.

Lines 212 – 215, 220 – 224, 242 – 243: the flow of writing need to be improved as it appears disjointed.

- According to a study by Cowan et al. in the US, a common reason for not receiving the vaccine was the lack of time , reported by 34% of participants. similarly, Wallace etal.found that 5-60% of health care workers cited time constraints as a reason for rejecting the influenza vaccine [17,26]. Among male primary care physicians, those who had received five or more influenza vaccinations in the past showed a significant positive association with self-efficacy. This finding aligns with the results of Alenazi et al., who found that increasing age, longer work duration in health services, being a male physician, are associated with significantly larger vaccination compliance (p=0.02, p=0.07, p=0.01, p=0.01) respectively [27]. The study revealed that physicians primarily rely on official sources for information about the influenza vaccine, such as the Ministry of Public Health ( 59.5) , WHO, and CDC (56.3), which is slightly lower than the 70.5% reported by AlMarzooqi et al.in 2018 [25]. In contrast, they consult electronic and print media and flu-specific resources much less frequently for influenza information. Alame et al. reported the participants relied on various sources for vaccine recommendations, including international organizations (e.g., WHO), and the Ministry of health [30].

References:

To check reference (15): the page in the link could not be found. It is not necessary to explain on what the reference is about in the reference list.

o 15. World Health Organization (WHO). (2020).* Influenza vaccination: An essential preventive measure. *WHO Vaccine Report*. Retrieved from https://www.who.int/influenza/vaccines/en/

Reviewer #2 :

We would like to express our sincere thanks to the reviewer for their thoughtful evaluation and helpful recommendations. The revisions made in response to these comments are marked in the updated manuscript and outlined in this rebuttal letter.

done

Reviewer #2: Dear Author,

Thank you for this study on very important topic of global concerns.

I think you still need a lot of work to put into it for it to fulFl the minimum

criteria of a good research article. Please Fnd below my comments:

Ln 1: I don't think this word (Assessing) is needed in the title.

You may leave it if you feel so strong about it.

Done

Ln 36 -57: While you have very important points in this abstract, it does not

have the minimum requirement of a good research abstract.

Please re-write this section bearing in mind the following:

-Introduction

-Methods

-Results

-Discussions

-straight to point and brief

-Word count

Note that your Abstract should be able to invite the readers to read the

whole of the study. Thanks

Done

Ln 66/67: Reference please.

Done

Ln 67: T

Done

Ln 67:is reinforced

Done

Ln 74/75: Reference please

done

Reviewer #2: Dear Author,

Thank you for this study on very important topic of global concerns.

I think you still need a lot of work to put into it for it to fulFl the minimum

criteria of a good research article. Please Fnd below my comments:

Ln 1: I don't think this word (Assessing) is needed in the title.

You may leave it if you feel so strong about it.

Done

Ln 36 -57: While you have very important points in this abstract, it does not

have the minimum requirement of a good research abstract.

Please re-write this section bearing in mind the following:

-Introduction

-Methods

-Results

-Discussions

-straight to point and brief

-Word count

Note that your Abstract should be able to invite the readers to read the

whole of the study. Thanks

Done

Ln 66/67: Reference please.

Done

Ln 67: T

Done

Ln 67:is reinforced

Done

Ln 74/75: Reference please

done

Reviewer #2: Dear Author,

Thank you for this study on very important topic of global concerns.

I think you still need a lot of work to put into it for it to fulFl the minimum

criteria of a good research article. Please Fnd below my comments:

Ln 1: I don't think this word (Assessing) is needed in the title.

You may leave it if you feel so strong about it.

Done

Ln 36 -57: While you have very important points in this abstract, it does not

have the minimum requirement of a good research abstract.

Please re-write this section bearing in mind the following:

-Introduction

-Methods

-Results

-Discussions

-straight to point and brief

-Word count

Note that your Abstract should be able to invite the readers to read the

whole of the study. Thanks

Done

Ln 66/67: Reference please.

Done

Ln 67: T

Done

Ln 67:is reinforced

Done

Ln 74/75: Reference pleas

Ln 1: I don't think this word (Assessing) is needed in the title.

You may leave it if you feel so strong about it.

o It was edited out.

Ln 36 -57: While you have very important points in this abstract, it does not have the minimum requirement of a good research abstract.

o It was rewritten

Ln 66/67: Reference please.

o Reference was added

Ln 67: T

o Done

Ln 67:is reinforced

o It has been edited

Ln 74/75: Reference please

o It has been added

Ln 77-79: Reference please

o It has been added

Ln 86: replace with "to"

o Done

Ln 88- 90: This is the statement of the scientific value of your study. You need to summarize the "hows?" in this introduction as well.

I will prefer the paragraph is framed as:

This research looked into improving attitudes and self-efficacy........... by:

1.Boosting the knowledge of the PCPs in training about influenza dx..

2. Ready availability of the influenza vax to pcp at their working units..

3...

4....

etc.

o “This research looked into improving attitude and self-efficacy of PCPs by designing strategies that boost their knowledge in training about influenza vaccine information and foster positive attitudes and strong self-efficacy through targeted messaging, healthcare provider involvement, enhanced accessibility, and availability of vaccine at their working u

---

## [Decision Letter · Decision Letter 1]

19 Oct 2025

PONE-D-25-15876R1Attitude, self-efficacy, and perceived risk toward seasonal influenza vaccination among primary care physicians in Qatar: A cross-sectional study.PLOS ONE

Dear Dr. Bibars,

Thank you for submitting your manuscript to PLOS ONE. After careful consideration, we feel that it has merit but does not fully meet PLOS ONE’s publication criteria as it currently stands. Therefore, we invite you to submit a revised version of the manuscript that addresses the points raised during the review process.

We look forward to receiving your revised manuscript.

Kind regards,

Ameel Al Shawi

Academic Editor

PLOS ONE

Journal Requirements:

Reviewers' comments:

Reviewer's Responses to Questions

**Comments to the Author**

1. If the authors have adequately addressed your comments raised in a previous round of review and you feel that this manuscript is now acceptable for publication, you may indicate that here to bypass the “Comments to the Author” section, enter your conflict of interest statement in the “Confidential to Editor” section, and submit your "Accept" recommendation.

Reviewer #1: (No Response)

Reviewer #2: All comments have been addressed

2. Is the manuscript technically sound, and do the data support the conclusions?

Reviewer #1: Yes

Reviewer #2: Yes

3. Has the statistical analysis been performed appropriately and rigorously? 

Reviewer #1: Yes

Reviewer #2: Yes

4. Have the authors made all data underlying the findings in their manuscript fully available?

Reviewer #1: No

Reviewer #2: Yes

5. Is the manuscript presented in an intelligible fashion and written in standard English?

Reviewer #1: Yes

Reviewer #2: Yes

6. Review Comments to the Author

Reviewer #1: General comments:

Thank you for giving me the opportunity to review the revised manuscript. The author had addressed all the comments. However, there are some comments for further improvement.

Specific comments as below:

Introduction:

Line 74-78: This study did not investigate strategies to improve PCPs’ attitude and self-efficacy; therefore, this sentence does not align with the study’s aims. Suggest remove or re-phrase the sentence.

Method:

Line 84-85: The word ‘physician’ is in lower case; ‘A list of physicians’ emails….’

Line 85: Write ‘PHCC’ in full at first mention and use the abbreviation from line 88 onwards.

Line 97-100: This sentence is lengthy and difficult to understand. Suggest rephrasing.

Line 102: To improve clarity, suggest amending: ‘….five-point Likert scale questions’.

Line 119: Is there a missing sentence after ‘The…’?

Line 128: The letter ‘a’ is lower case in this sentence; ….a survey link…

Results:

Line 146: the term ‘primary care physician’ was earlier used, suggest to standardize throughout the manuscript.

Formatting of Table 2 can be improved.

Table 4: the first column described the ‘statements’. ‘Those who answered ‘agree’ and ‘strongly agree’ are the description for the second and third columns (‘N’ and ‘%’).

Limitations of the study:

Line 260: The limitation of this study was the low response rate.

Suggestion for the author to check the overall manuscript as there are some spelling mistakes to be corrected e.g. ‘health canter’ and to check the use of proper noun and common noun.

Reviewer #2: Dear Author,

Thank you for addressing my concerns appropriately in this round of reviewing.

I think your study is very relevant in preventive health care

I wish you the best outcome.

7. PLOS authors have the option to publish the peer review history of their article (what does this mean? ). If published, this will include your full peer review and any attached files.

**Do you want your identity to be public for this peer review?** For information about this choice, including consent withdrawal, please see our Privacy Policy .

Reviewer #1: No

Reviewer #2: No

---

## [Author Response · Author response to Decision Letter 2]

1 Dec 2025

Reviewers' comments:

Reviewer's Responses to Questions

Comments to the Author

1. If the authors have adequately addressed your comments raised in a previous round of review and you feel that this manuscript is now acceptable for publication, you may indicate that here to bypass the “Comments to the Author” section, enter your conflict of interest statement in the “Confidential to Editor” section, and submit your "Accept" recommendation.

Reviewer #1: (No Response)

Reviewer #2: All comments have been addressed

2. Is the manuscript technically sound, and do the data support the conclusions?

Reviewer #1: Yes

Reviewer #2: Yes

3. Has the statistical analysis been performed appropriately and rigorously?

Reviewer #1: Yes

Reviewer #2: Yes

4. Have the authors made all data underlying the findings in their manuscript fully available?

Reviewer #1: No

Reviewer #2: Yes

An official email was sent to primary health care clinical research department and this was the response: Data underlying the findings are subject to PHCC Qatar’s data sharing policy and IRB restrictions. Aggregate data supporting the results are available within the manuscript. Individual-level data cannot be publicly shared.

5. Is the manuscript presented in an intelligible fashion and written in standard English?

Reviewer #1: Yes

Reviewer #2: Yes

6. Review Comments to the Author

Reviewer #1: General comments:

Thank you for giving me the opportunity to review the revised manuscript. The author had addressed all the comments. However, there are some comments for further improvement.

Specific comments as below:

Introduction:

Line 74-78: This study did not investigate strategies to improve PCPs’ attitude and self-efficacy; therefore, this sentence does not align with the study’s aims. Suggest remove or re-phrase the sentence.

-removed

Method:

Line 84-85: The word ‘physician’ is in lower case; ‘A list of physicians’ emails….’

-Done

Line 85: Write ‘PHCC’ in full at first mention and use the abbreviation from line 88 onwards.

-Done

Line 97-100: This sentence is lengthy and difficult to understand. Suggest rephrasing.

-A validated questionnaire from Asma et al. [16] was adapted with minor demographic changes, including PHCC work experience. The second section assessed attitude using 13 items, of which 10 measured self-efficacy, five on social influence and five on personal competence.

Line 102: To improve clarity, suggest amending: ‘….five-point Likert scale questions’.

-Done

Line 119: Is there a missing sentence after ‘The…’?

-Removed

Line 128: The letter ‘a’ is lower case in this sentence; ….a survey link…

-Done

Results:

Line 146: the term ‘primary care physician’ was earlier used, suggest to standardize throughout the manuscript.

-Done

Formatting of Table 2 can be improved.

-Done

Table 4: the first column described the ‘statements’. ‘Those who answered ‘agree’ and ‘strongly agree’ are the description for the second and third columns (‘N’ and ‘%’).

Done

Limitations of the study:

Line 260: The limitation of this study was the low response rate.

The low response rate and small sample size may restrict generalizability. Vaccination uptake was self-reported rather than verified with official records, introducing potential social desirability bias. Recall bias is also possible; however, the survey was conducted over three weeks during the flu season (October 2021).

Suggestion for the author to check the overall manuscript as there are some spelling mistakes to be corrected e.g. ‘health canter’ and to check the use of proper noun and common noun.

Done

Reviewer #2: Dear Author,

Thank you for addressing my concerns appropriately in this round of reviewing.

I think your study is very relevant in preventive health care

I wish you the best outcome.

7. PLOS authors have the option to publish the peer review history of their article (what does this mean?). If published, this will include your full peer review and any attached files.

Do you want your identity to be public for this peer review? For information about this choice, including consent withdrawal, please see our Privacy Policy.

Reviewer #1: No

Reviewer #2: No

---

## [Editor Report · Decision Letter 2]

16 Dec 2025

Attitude, self-efficacy, and perceived risk toward seasonal influenza vaccination among primary care physicians in Qatar: A cross-sectional study.

PONE-D-25-15876R2

Dear Dr. Bibars,

We’re pleased to inform you that your manuscript has been judged scientifically suitable for publication and will be formally accepted for publication once it meets all outstanding technical requirements.

Kind regards,

Ameel Al Shawi

Academic Editor

PLOS One
---

## [Editor Report · Acceptance letter]

PONE-D-25-15876R2

PLOS One

Dear Dr. Bibars,

I'm pleased to inform you that your manuscript has been deemed suitable for publication in PLOS One. Congratulations! Your manuscript is now being handed over to our production team.

Kind regards,

on behalf of

Dr. Ameel Al Shawi

Academic Editor

PLOS One